# MIXING CONFIGURATIONS FOR DOWNSTREAM PREDICTION

## ABSTRACT

Humans possess an innate ability to group objects by similarity—a cognitive mechanism that clustering algorithms aim to emulate. Recent advances in community detection have enabled the discovery of *configurations*—valid hierarchical clusterings across multiple resolution scales—without requiring labeled data. In this paper, we formally characterize these configurations and identify similar emergent structures in register tokens within Vision Transformers. Unlike register tokens, configurations exhibit lower redundancy and eliminate the need for ad hoc selection. They can be learned through unsupervised or self-supervised methods, yet their selection or composition remains specific to the downstream task and input. Building on these insights, we introduce GraMixC, a plug-and-play module that extracts configurations, aligns them using our novel Reverse Merge/Split (RMS) technique, and fuses them via attention heads before forwarding them to any downstream predictor. On the DSNI 16S rRNA cultivation-media prediction task, GraMixC improves the $R^2$ from 0.6 to 0.9 on various methods, setting a new state-of-the-art. We further validate GraMixC across standard tabular benchmarks, where it consistently outperforms single-resolution and static-feature baselines.

## 1 INTRODUCTION

Learning general-purpose features that enhance downstream tasks has been a long-standing goal in machine learning. One prominent example is clustering (*i.e.*, community detection) in unsupervised learning, which groups entities into clusters of similar objects while separating dissimilar ones, without using labels (MacQueen, 1967; Jianbo Shi & Malik, 2000; Ng et al., 2001). Interestingly, this paradigm demonstrates remarkable similarities to human-like behaviors. Decades of cognitive science studies show that even infants have the ability to group objects by similarity (Quinn & Eimas, 1996; Bornstein et al., 2010). In particular, they often organize them at different abstraction levels (Zaadnoordijk et al., 2022; Muttenthaler et al., 2024). Inspired by this, recent advances in community detection have extended clustering to the discovery of *configurations*—hierarchical clusterings that span multiple resolution scales (Pitsianis et al., 2023). For example, as illustrated in the lineage diagram of Fig. 1, in the CIFAR10 dataset (Krizhevsky, 2009), coarse configurations may separate vehicles from animals, while finer configurations distinguish between birds, cats, and dogs. These multi-resolution representations reveal rich hierarchical structures that could provide stronger priors or inductive biases for deep models. However, despite their potential, such configurations remain largely underexplored in deep learning, especially in challenging domains where labels are sparse.

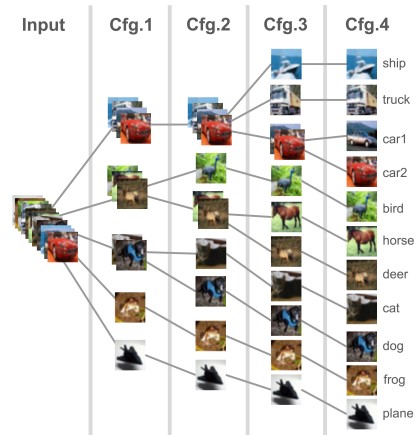

Figure 1: Illustration of CIFAR10 configurations. Each column represents a configuration—clustering at a specific resolution.

One such domain is 16S ribosomal RNA (rRNA) gene sequencing, a widely used tool in microbiome studies for identifying and classifying bacteria. Analyzing 16S rRNA data has consistently confronted significant challenges in downstream prediction tasks within label-scarce environments.

Previous works in 16S rRNA representation learning have demonstrated substantial benefits for bacterial taxonomic profiling and microbial community analysis (Janda & Abbott, 2007; Wang et al., 2007; De Vrieze et al., 2018). Notably, Johnson et al. (2019) showed that full-length sequencing combined with appropriate clustering of intragenomic sequence variation can provide more accurate representation of bacterial species in microbiome datasets. These findings underscore the importance of learning clustered representations without relying on labels.

Recent methodologies typically transform clustering results into pseudo-labels to enhance downstream prediction performance. For instance, DeepCluster (Caron et al., 2019) iteratively clusters CNN-extracted visual features and leverages these cluster assignments to guide network parameter updates. Graph-based methods such as (Yang et al., 2023) employ structural clustering to overcome limitations of traditional contrastive learning approaches that depend on positive and negative sample pairs. Their method captures structural relationships among nodes in heterogeneous information networks, establishing a self-supervised pre-training framework that learns robust network representations from unlabeled data. Nevertheless, these approaches predominantly focus on a single configuration, overlooking the potential benefits of mixing configurations across multiple resolution scales.

In this paper, we introduce GraMixC, a plug-and-play module that extracts, aligns and mixes graph-based configurations for downstream prediction. The main contributions of the paper are as follows:

- We identify three key characteristics of clustering configurations through systematic experimental analysis, providing a novel perspective on enhancing downstream prediction via mixing configurations.
- We propose GraMixC, a plug-and-play module based on mixed configurations. We apply it to a novel 16S rRNA cultivation-media prediction task, setting a new state-of-the-art.
- We further conduct extensive experiments on multiple standard tabular benchmarks to validate GraMixC's effectiveness, where it consistently outperforms single-resolution and static-feature baselines.

The remainder of this paper is organized as follows. Section 2 analyzes behavioral patterns of configurations. Section 3 details our proposed GraMixC. Section 4 evaluates GraMixC's performance through extensive experiments. Finally, Section 5 concludes the paper. Our data and implementation is available at `https://anonymous.4open.science/r/project-34CB`.

## 2 PRELIMINARY RESULTS

We first present preliminary experimental results on configurations using CIFAR10. Specifically, we compare patterns of configurations with those of the learnable "register" tokens in a recent vision transformer DINOv2-reg (Darcet et al., 2024). Fig. 2 shows the attention maps from our configurations and their register tokens. Moreover, Fig. 3 shows qualitative behaviors of our configurations and their quantitative advantages over registers in terms of feature importance and neighborhood similarity. From these results, we identify three key properties:

**Configurations emerge via unsupervised or self-supervised learning.** We define Near ground truth (GT) balls as balls selected with the highest clustering scores, marked yellow in Fig. 2a. As shown in Fig. 2b, the attention map, acquired by feeding configurations as tokens to attention heads for linear probing, yields high norm regions substantially overlap with GT balls. On another hand, DINOv2-reg exhibits similar attention map patterns in selected registers (see Fig. 2c), which might be related to registers activating different areas in Fig. 2d, similar to slot attention (Locatello et al., 2020; Caron et al., 2021; Oquab et al., 2024; Darcet et al., 2024). Thus, based on the similar attention map behavior, register token can be considered as a latent configuration.

**Configurations are selected and mixed based on input and task.** *Configuration selection and mixing* refers to learning which resolution scales to focus on for a given downstream task. We visualize this via attention maps over configuration tokens, where high-norm regions indicate the selected scales. In Fig. 2b, attention norms vary across rows, showing that each input sample triggers different resolution scales. Without any change to the configurations, we merge the original labels into coarser classes (Fig. 3a) and plot the new attention map (Fig. 3b). The attention shifts to align with the coarser GT, whereas DINOv2-reg register tokens remain unchanged unless re-trained. These observations confirm that configuration selection and mixing are input- and task-dependent.

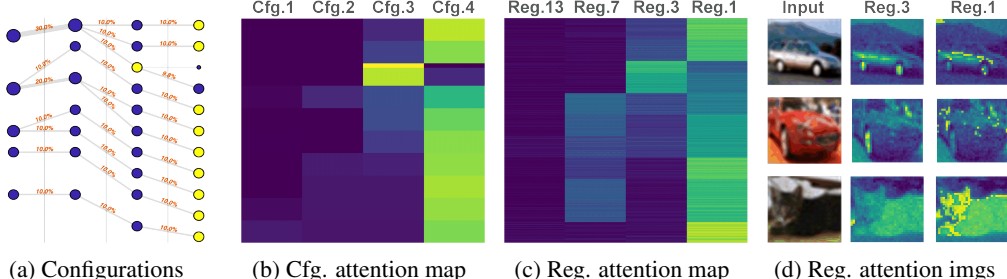

(a) Configurations     (b) Cfg. attention map     (c) Reg. attention map     (d) Reg. attention imgs

Figure 2: Comparison of attention maps obtained from configurations and registers, rows for samples. **(a)**: Lineage diagram for configurations, near GT balls are marked yellow. **(b)**: Attention map of configuration tokens in an attention-based linear probing. **(c)**: Attention map of DINOv2-reg register tokens, mean of all patch norms is used. **(d)**: Attention maps over the register tokens, as images.

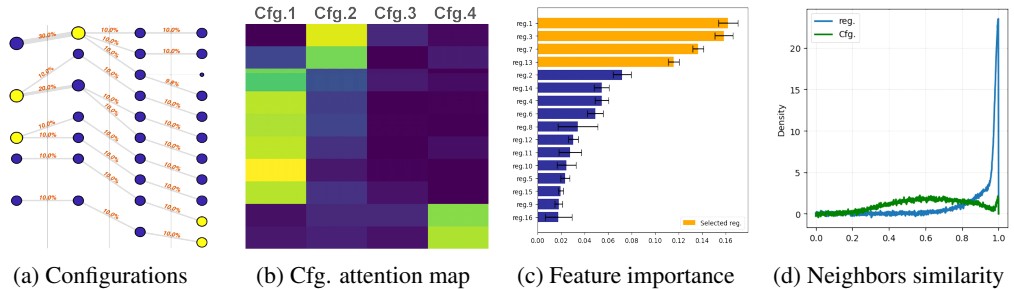

(a) Configurations     (b) Cfg. attention map     (c) Feature importance     (d) Neighbors similarity

Figure 3: Illustration of another two properties of configurations, grouped by left two and right two. **(a)**: Lineage diagram where coarser classes are used for GT. **(b)**: Attention map in linear probing the coarser classes. **(c)**: Distribution of feature vector importance over the register tokens querying, mean of all patch importance is used. **(d)**: Distribution of cosine similarity between query embeddings of register and configuration tokens and their 2 neighbors, mean of all patch similarities is used.

**Configurations are more informative and less redundant than register tokens.** Register tokens can help extract configurations, similar to object detection (Siméoni et al., 2021; Zhang et al., 2022), but selecting a fixed number by feature importance is arbitrary and non-rigorous (see Fig. 3c). Furthermore, register tokens exhibit high redundancy—cosine similarity between their embeddings and their 2 neighbors embeddings is heavily skewed toward 1—whereas configurations yield information less redundant (see Fig. 3d).

## 3 METHODOLOGY

Having these characterizations, we hypothesize that unsupervised methods can produce hierarchical *multi-resolution clusterings*, and that task- and input-specific *selection and mixing* of these configurations represent *global information* beneficial to downstream tasks. Building on the hypothesis, we propose a lightweight module *GraMixC*, that treats configurations as tokens ([CFG]) and incorporates a novel alignment layer plus learnable attention heads (Vaswani et al., 2017) after the configuration extraction model, enabling task- and input-specific mixing of configurations via end-to-end back-propagation.

Fig. 4 illustrates GraMixC. Given an input matrix $\boldsymbol{X} \in \mathbb{R}^{N \times d}$ (with $N$ samples and feature dimension $d$), GraMixC pass $\boldsymbol{X}$ to two branches: (1) a path to unsupervised learning box that extracts configurations, and (2) a direct path to the downstream predictor. If at inference, we apply *Reverse Merge & Split* (RMS) alignment on the configurations. Then we pass them to positional encoding (PE) and attention heads. The final concatenation is passed to a downstream predictor for the prediction $\tilde{\boldsymbol{y}}$.

Except for the downstream predictor, the GraMixC model can be divided into three parts: unsupervised configuration learning, the Reverse Merge & Split (RMS) for alignment, and attention heads for fusion. In the attention heads part, following Darcet et al. (2024), we append register tokens [REG]

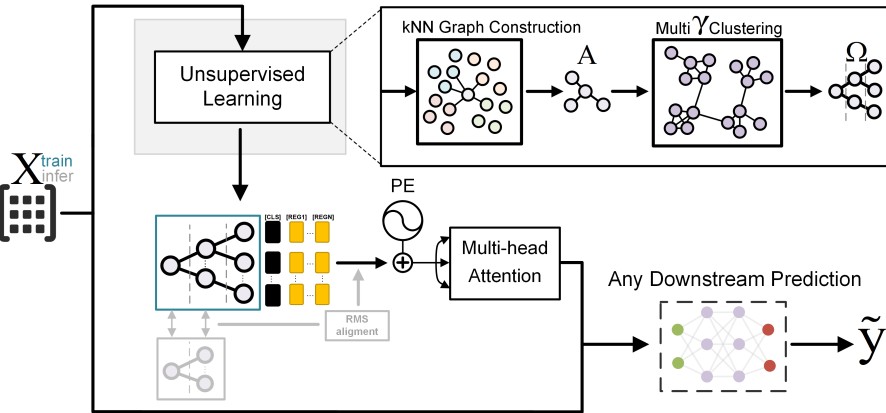

Figure 4: Illustration of the proposed GraMixC module and resulting model. The input data branches into (upper) a path to unsupervised learning box that extracts configurations, and (lower) a direct path to the downstream predictor. Their outcomes concatenate and pass to the downstream predictor. The components occur only during training and inference are colored in blue and gray, respectively.

after `[CFG]` and `[CLS]` for a clean attention map, that can be used backwards to guide configuration selection. Below we detail the rest two components in Section 3.1 and Section 3.2.

### 3.1 MULTI-RESOLUTION GRAPH-BASED CLUSTERING

Given $\boldsymbol{X}$, multi-resolution clustering seeks to extract *configurations*—valid hierarchical clusterings across multiple resolution scales—which we denote as $\boldsymbol{\Omega} \in \mathbb{N}^{N \times m}$, where $m$ denotes the number of valid resolution levels. To preserve the latent manifold structure in data, ease parameter sensitivity, and prevent other problems with traditional clustering methods (see Section D), we choose the resolution parameter ($\gamma \in \mathbb{R}_+$)-based community detection as our core clustering method. While BlueRed (Liu et al., 2021) can conduct graph clustering without problems like resolution limit or parameter sensitivity in traditional methods, recent work by Pitsianis et al. (2023) further demonstrates the elimination of $\gamma$ selection, and enabled the unsupervised discovery of $\boldsymbol{\Omega}$ and the corresponding set of all valid $\gamma$, which is denoted as $\Gamma = \{\gamma_1^*, \gamma_2^*, \ldots, \gamma_m^*\} \subseteq [0, \infty)$. Inspired by these works, the unsupervised box in Fig. 4 unfolds into two steps: **(1) k-nearest neighbors (kNN)** (Tenenbaum et al., 2000) **graph construction**, which return a directed graph $G = (V, E)$, usually represented as adjacency matrix $\boldsymbol{A} \in \mathbb{R}_+^{N \times N}$, and **(2) multi-$\gamma$ clustering** on the resulted graph, *i.e.* modularity based community detection with unsupervised $\Gamma$ learning, which return the wanted $\boldsymbol{\Omega}$. The details for each of these two steps are:

**(1) kNN graph construction.** We construct a kNN graph with $k = \log_{10} N$ as convention, using Euclidean distance for simplicity. Such pair-wise geometric distance between two different vertexes is denoted $d(\boldsymbol{x}_i, \boldsymbol{x}_j)$ where $i \neq j$ and $x_i \in \mathbb{R}^d$ is the $i$-th feature vector. We then have the adjacency matrix $\boldsymbol{A}$ formulated as: $A_{ij} = d(\boldsymbol{x}_i, \boldsymbol{x}_j)$ if $(\boldsymbol{x}_i, \boldsymbol{x}_j) \in E$, 0 otherwise, where $E$ is the edge set of the kNN graph and $A_{ij}$ denotes the $i$-th row and $j$-th column element of the adjacency matrix. Then we force *column stochastic* by dividing each column in the constructed $\boldsymbol{A}$ with the column sum. The resulted graph is sparse stochastic, and we can apply Stochastic Graph t-SNE (SG-t-SNE) reweighting (Pitsianis et al., 2019), which proved to remedy skewed degree distribution, that is not promised by conventional t-SNE (Van der Maaten & Hinton, 2008). From the original work, the key equations for SG-t-SNE reweighting are:

$$w(\boldsymbol{x}_i, \boldsymbol{x}_j) = \frac{1}{\lambda} \exp\left(-\frac{d^2(\boldsymbol{x}_i, \boldsymbol{x}_j)}{2\sigma_i^2}\right), \quad \text{with} \quad \lambda = \sum_{\boldsymbol{x}_j : (\boldsymbol{x}_i, \boldsymbol{x}_j) \in E} \exp\left(-\frac{d^2(\boldsymbol{x}_i, \boldsymbol{x}_j)}{2\sigma_i^2}\right),$$

where $\lambda$ is a non-negative constant, which we simply set to 15 as previous work show that it is not so sensitive to the choice of $\lambda$ (Pitsianis et al., 2019), and $\sigma_i$ is a variable to be numerically solved with bisection method. After giving value of $w$ to $d$, we have $\boldsymbol{A}$ with less skewed degree distribution, which avoids problems like numerical instability and bias towards hubs in downstream clustering.

**(2) multi-$\gamma$ community detection.** Then one may simply pass the reweighted $\boldsymbol{A}$ to $\gamma$-based community detection method, such as Leiden algorithm (Traag et al., 2019), to get one pseudo-configuration vector $\boldsymbol{\omega}_\gamma \in \{1, \ldots, N\}^N$ ("pseudo" for not sure to be valid). However, such $\gamma$ falls in the range of $[0, \infty)$, and searching over all possible $\gamma$ is exhausting. Therefore, we incorporate the BlueRed method with parallel descending triangulation (parallel-DT) (Pitsianis et al., 2023), in order to automatically discover all valid $\gamma^* \in \Gamma$. Given a fixed $\gamma$, BlueRed find the optimal configuration $\boldsymbol{\omega}_\gamma$ by the following optimization:

$$\boldsymbol{\omega}_\gamma = \underset{\boldsymbol{\omega} \in \{1, \ldots, N\}^N}{\arg\min} \left[ -\sum_{k=1}^{|\boldsymbol{\omega}|_\infty} \sum_{(i,j) \in E} d(\boldsymbol{x}_i, \boldsymbol{x}_j) \mathbf{1}_{\omega_i = \omega_j = k} + \gamma \sum_{k=1}^{|\boldsymbol{\omega}|_\infty} \sum_{(i,j) \in E} d^2(\boldsymbol{x}_i, \boldsymbol{x}_j) \mathbf{1}_{\omega_i = k}, \right],$$

where $\omega_i$ denotes the $i$-th element of $\boldsymbol{\omega}$, $|\boldsymbol{\omega}|_\infty = \max_{i \leq N} \omega_i$ is a inf-norm, and $\mathbf{1}$ denotes the indicator gate which take value 1 if its subscript condition holds, 0 otherwise. Pitsianis et al. (2023) describe the first term as attraction and the second term as repulsion. Optimizing each solely yields all-in-one configuration $\boldsymbol{\omega}_0 = [1, 1, \ldots, 1]$ and all-lonely configuration $\boldsymbol{\omega}_\infty = [1, 2, \ldots, N]$. Between these two configurations, parallel-DT allows forming BlueRed Front (BRF) (Pitsianis et al., 2023) by segmenting $(0, \infty)$ into $m$ ranges, among which each has a dominant $\gamma_i^*$ yields lower HAR (Pitsianis et al., 2023)—the sum of first term and the negative second term—which means "local minimum" on that range. Thus desired $\boldsymbol{\Omega}$ is formed.

## 3.2 RMS: REVERSE MERGE & SPLIT ALIGNMENT

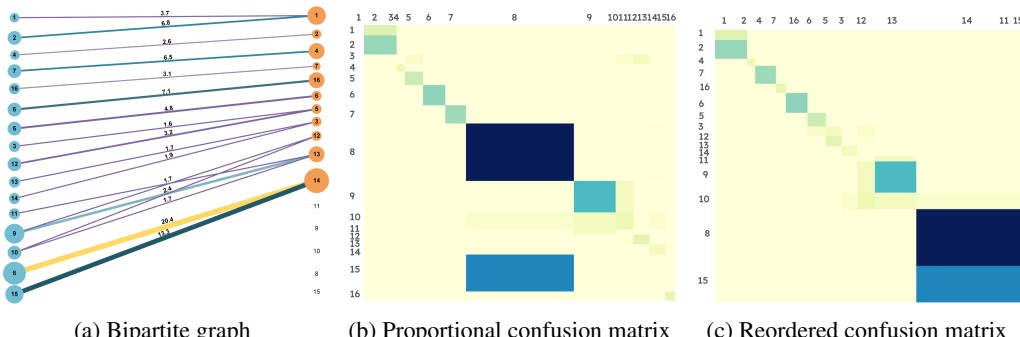

(a) Bipartite graph      (b) Proportional confusion matrix      (c) Reordered confusion matrix

Figure 5: Example of the RMS alignment process applied to clustering results and ground truth (both treated as configurations) on the Salinas dataset (Plaza & Tilton, 2005). **(a)**: Bipartite graph representation, where blue nodes correspond to predicted clusters and red nodes to ground truth clusters. Node labels indicate cluster indices; edge labels show the proportion of samples shared between clusters. **(b)**: Proportional confusion matrix $\boldsymbol{C}$ comparing predicted clusters (horizontal axis) to ground truth clusters (vertical axis). **(c)**: Confusion matrix $\boldsymbol{C}_{\text{tw}}$ reordered via the two-walk Laplacian. Notable splits, such as ground truth cluster 8 being divided into clusters 8 and 15 in the prediction, can be resolved through the reverse merge/split procedure.

Multi-resolution clustering on different datasets $\boldsymbol{X}_{\text{train}}$ and $\boldsymbol{X}_{\text{test}}$ often naturally produces misaligned configurations, that either (1) have different value of $m$ or $|\boldsymbol{\omega}|_\infty$, or (2) have different cluster labels. While (2) is not a problem as re-assigning fix it, (1) could be problematic as the length and position of configurations influence the downstream fusion. One possible interpretation is that some clusters are further merged or split in another configuration, leading to this mismatch. To address this, we propose Reverse Merge & Split (RMS), which identifies an optimal alignment, allowing re-merging and re-splitting, between two configurations, $\boldsymbol{\omega}_i$ and $\boldsymbol{\omega}_j$. First of all, an alignment score is defined:

$$\text{SCORE}(\boldsymbol{\omega}_i, \boldsymbol{\omega}_j) = \text{ARI}(\boldsymbol{\omega}_i, \boldsymbol{\omega}_j) - \theta \left| \frac{|\boldsymbol{\omega}_i|_\infty - |\boldsymbol{\omega}_j|_\infty}{|\boldsymbol{\omega}_i|_\infty + |\boldsymbol{\omega}_j|_\infty} \right|.$$

where $\theta$ is a hyperparameter to balance the weights of the two terms, which we set to 0.1, ARI is the adjusted rand index as defined in Hubert & Arabie (1985). By this punished ARI design, we consider different labels, merge and split during scoring the alignment between two partition, but also avoids too much difference in number of clusters (one extreme case is $\boldsymbol{\omega}_0$ and $\boldsymbol{\omega}_\infty$ has ARI of 1).

However, the SCORE itself does not convey the mapping we need for reassigning. In *RMS* alignment, we construct a confusion matrix $C \in \mathbb{N}^{|\omega_i|_\infty \times |\omega_j|_\infty}$ between $\omega_i$ and $\omega_j$. Fig. 5 illustrates this process with a concrete example, showing how the confusion matrix captures the relationship between predicted and ground truth clusters, including cases where clusters are split or merged across configurations. As an assignment problem with a rectangle cost matrix $-C$[1], it is solvable by twisting existing Hungarian algorithm methods (Kuhn, 1955; Jonker & Volgenant, 1987; Bertsekas, 1992). Because $C$ *is the adjacency matrix of a bipartite graph*, spectral reordering via its graph Laplacian is preferred, since it encodes global connectivity and reveals coherent split–merge structures rather than merely optimizing diagonal entries. As the Fiedler vector reordering (Fiedler, 1973) assumes symmetric positive semi-definite, it is not directly applicable to $C$. Inspired by a recent work of Floros et al. (2024), we introduce a *two-walk Laplacian*, which is defined as:

$$L_{\text{tw}} = D - C_{\text{tw}}, \quad \text{with} \quad C_{\text{tw}} = \begin{bmatrix} CC^\top & C \\ C^\top & C^\top C \end{bmatrix},$$

where $D = \text{diag}(C_{\text{tw}}\mathbf{1})$ is the diagonal degree matrix of $C_{\text{tw}}$. We remap $\omega_i$ and $\omega_j$ by using, respectively, the first $\|\omega_i\|_\infty$ and the last $\|\omega_j\|_\infty$ entries in the Fiedler eigenvector of $L_{\text{tw}}$, which is the eigenvector corresponds to smallest positive eigenvalue. We further reverse split and merge simply by reassigning the redundant columns or rows who has element larger than its diagonal entry.

In GraMixC, we carry a small portion (0.1%) of train samples as *anchors* during inference, and the portion of $\Omega_{\text{train}}$ and $\Omega_{\text{test}}$ corresponding to the anchors are used to calculate the SCORE. Given $m$ is usually small, we exhaustively test pairs $(\omega_i, \omega_j)$ then iteratively pick the pair yielding the highest SCORE for each $\omega_i$. For each pair, we apply the mapping from $\text{RMS}(\omega_i, \omega_j)$. The final alignments is then used to match the configurations. See our GitHub repository [2] and Section E for alignment examples and more implementation details.

## 4 EXPERIMENTS

In this section, we evaluate the proposed plug-and-play module by training baseline models with and without GraMixC (*GMC*). We also test a static variant (*GC*), which use aligned configurations as extra features, without attention mechanism. We expect the performance to follow a general trend

$$\text{baseline} < \text{baseline+GC} < \text{baseline+GMC}.$$

We then ablate the number of configurations used to check that they cause a performance regression.

### 4.1 IMPLEMENTATION DETAILS AND EXPERIMENTAL SETUP

Our module was implemented with MATLAB, Python 3.12, PyTorch 2.6. We run trainings on a GeForce RTX 3090Ti GPU. Models were trained with the Adam optimizer (Kingma & Ba, 2017) at a fixed learning rate of $10^{-3}$. Unless otherwise noted, we used a batch size of 100 and trained for up to 100 epochs.

Ahead of diving into the experimental details, we briefly summarize the datasets and metrics used.

**DSNI-pH and DSNI-Temp.** We collected the DSNI dataset from DSMZ (German Collection of Microorganisms and Cell Cultures GmbH, 2025) and NIH. It comprises six relational tables (STRAINS, MEDIA, SOLUTIONS, INGREDIENTS, STEPS, GAS) covering taxonomic and protocol information. We use approximately $65\,000$ samples with 16S rRNA sequence (500–1 500 nucleotides), cultivation temperatures (2–103 °C), and pH (0.1–11). The task is to predict optimal temperature (DSNI-Temp) and pH (DSNI-pH) from the 16S rRNA sequence.

Following Çelikkanat et al. (2024) and related works (Wood & Salzberg, 2014; Compeau et al., 2011), we encode each 16S rRNA sequence as a 7-mer count vector in $\mathbb{N}^{16\,384}$, yielding a dataset of shape $65\,023 \times 16\,384$. We perform an 80/20 split (52,018 train / 13,005 test), which preserves the skewed pH (6–8) and temperature (20–40 °C) distributions. Section C provides an illustration for target value ($y_{\text{train}}$ and $y_{\text{test}}$) distribution. Preprocessing—robust scaling, variance thresholding, and selection

---

[1]The negative of the confusion matrix is used to frame the assignment problem (minimizing the diagonal).
[2]https://anonymous.4open.science/r/project-82CE

Table 1: Regression performance on DSNI-pH, DSNI-Temp and QM9. Values are mean±std from runs with different random seeds; best results per baseline are bold; best results per metric are underlined.

| | DSNI-pH | | DSNI-Temp | | QM9 | |
|---|---|---|---|---|---|---|
| | MSE ↓ | $R^2$ | MSE ↓ | $R^2$ | MAE ↓ | $R^2$ |
| RF | $0.198_{\pm0.000}$ | $0.601_{\pm0.001}$ | $17.759_{\pm0.276}$ | $0.393_{\pm0.009}$ | $0.015_{\pm0.000}$ | $0.979_{\pm0.000}$ |
| XGBoost | $0.196_{\pm0.001}$ | $0.604_{\pm0.003}$ | $18.212_{\pm0.543}$ | $0.377_{\pm0.018}$ | $0.014_{\pm0.001}$ | $0.978_{\pm0.001}$ |
| CatBoost | $0.193_{\pm0.001}$ | $0.610_{\pm0.002}$ | $17.375_{\pm0.398}$ | $0.406_{\pm0.013}$ | $0.014_{\pm0.000}$ | $0.978_{\pm0.002}$ |
| 3LP | $0.201_{\pm0.002}$ | $0.595_{\pm0.006}$ | $18.484_{\pm0.183}$ | $0.368_{\pm0.006}$ | $0.018_{\pm0.001}$ | $0.958_{\pm0.001}$ |
| 3LP+GC | $0.097_{\pm0.004}$ | $0.804_{\pm0.008}$ | $6.520_{\pm0.360}$ | $0.777_{\pm0.012}$ | $0.016_{\pm0.003}$ | $0.974_{\pm0.000}$ |
| 3LP+GMC | $\mathbf{0.023}_{\pm0.002}$ | $\mathbf{0.953}_{\pm0.004}$ | $\mathbf{2.277}_{\pm0.061}$ | $\mathbf{0.922}_{\pm0.002}$ | $\mathbf{0.010}_{\pm0.003}$ | $\mathbf{0.990}_{\pm0.002}$ |
| TabN | $0.184_{\pm0.004}$ | $0.629_{\pm0.007}$ | $13.290_{\pm0.244}$ | $0.545_{\pm0.008}$ | $0.015_{\pm0.001}$ | $0.962_{\pm0.002}$ |
| TabN+GC | $0.086_{\pm0.003}$ | $0.825_{\pm0.007}$ | $7.997_{\pm0.210}$ | $0.726_{\pm0.007}$ | $0.012_{\pm0.002}$ | $0.983_{\pm0.001}$ |
| TabN+GMC | $\mathbf{0.020}_{\pm0.001}$ | $\mathbf{0.959}_{\pm0.002}$ | $\underline{\mathbf{0.989}}_{\pm0.361}$ | $\mathbf{0.966}_{\pm0.012}$ | $\underline{\mathbf{0.008}}_{\pm0.000}$ | $\mathbf{0.995}_{\pm0.002}$ |
| TabT | $0.256_{\pm0.007}$ | $0.483_{\pm0.014}$ | $18.910_{\pm0.247}$ | $0.353_{\pm0.008}$ | $0.434_{\pm0.008}$ | $0.921_{\pm0.008}$ |
| TabT+GC | $0.106_{\pm0.002}$ | $0.786_{\pm0.005}$ | $8.280_{\pm0.303}$ | $0.717_{\pm0.010}$ | $0.212_{\pm0.004}$ | $0.961_{\pm0.008}$ |
| TabT+GMC | $\mathbf{0.017}_{\pm0.002}$ | $\mathbf{0.964}_{\pm0.005}$ | $\mathbf{2.785}_{\pm0.540}$ | $\mathbf{0.904}_{\pm0.018}$ | $\mathbf{0.009}_{\pm0.000}$ | $\underline{\mathbf{0.998}}_{\pm0.001}$ |
| FTT | $0.218_{\pm0.003}$ | $0.561_{\pm0.006}$ | $13.571_{\pm0.069}$ | $0.536_{\pm0.002}$ | $0.085_{\pm0.005}$ | $0.984_{\pm0.006}$ |
| FTT+GC | $0.070_{\pm0.003}$ | $0.858_{\pm0.007}$ | $5.915_{\pm0.277}$ | $0.797_{\pm0.009}$ | $0.034_{\pm0.002}$ | $0.993_{\pm0.003}$ |
| FTT+GMC | $\underline{\mathbf{0.007}}_{\pm0.005}$ | $\underline{\mathbf{0.984}}_{\pm0.009}$ | $\mathbf{1.480}_{\pm0.120}$ | $\mathbf{0.949}_{\pm0.004}$ | $\mathbf{0.026}_{\pm0.001}$ | $\mathbf{0.995}_{\pm0.003}$ |

of the top 1,000 features—was fitted on the training set and then applied to both splits to avoid data leakage.

**Additional benchmarks.** We further evaluate on QM9 (Ramakrishnan et al., 2014) for molecular property regression, on Boston Housing (Harrison & Rubinfeld, 1978), and on MNIST (Lecun et al., 1998) and CIFAR10 for classification (some in Section F).

**Evaluation metrics.** For regression we use mean squared error (MSE), mean absolute error (MAE; used for QM9 for comparability with SOTA) for training, and report coefficient of determination ($R^2$). For classification we use cross-entropy loss (CE) for training and report top-1 accuracy (Acc).

For each benchmark, we include three classical decision tree models for reference: Random Forest (RF) (Breiman, 2001), XGBoost (Chen & Guestrin, 2016), CatBoost (Prokhorenkova et al., 2018). As both GMC and GC are plug-and-play modules, they can be easily applied to various downstream predictors. We first evaluate a 3-layer perceptron (3LP) with hidden dims [256,128,64]. Because our inputs combine numerical features with categorical configurations, we naturally consider tabular models: TabNet (TabN) (Arik & Pfister, 2020), TabTransformer (TabT) (Huang et al., 2020), FT-Transformer (FTT) (Gorishniy et al., 2023) were all run with their default settings from the official implementations.

## 4.2 EVALUATION OF THE PROPOSED MODULE

As shown in Fig. 2 and Fig. 3, we demonstrate, with attention maps, the learned mixing of configurations by training models with self-attention head on aligned configurations. In order to quantify the quality of such mixing, for each baseline, we set up the evaluation in three modes: standalone (baseline), with static configuration concatenation (baseline+GC), and with attention-based fusion via GraMixC (baseline+GMC). Table 1 reports regression results on our main benchmarks; Section F (Table 2) shows the rest results. Across all models and tasks, adding GC yields consistent gains, and incorporating GMC provides further significant improvements, confirming our initial hypothesis.

**Performance improvement.** Table 1 shows that adding GC and GMC yields consistent gains across all baselines. Among these observed improvements, the scores increasing on DSNI is quite satisfying. Prior specialized growth-media regression methods are not convincing with $R^2 \leq 0.8$ (e.g., 0.75 (Sauer & Wang, 2019)). We confirm this with our base models score $R^2$ between 0.3 and 0.6 on DSNI-pH and DSNI-Temp. However, even without tailoring the baseline model design, we bring the score to a new high by simply adding GC or GMC. Fig. 6 illustrates some examples

of such improvement. We see the model's predictions align more closely with the ideal regression line and better handle rare cases, by incorporating configurations and probably capturing the latent manifold structure. Incorporating GC and further GMC raises $R^2$ to 0.98 (pH) and 0.97 (Temp). Which not only is considered very satisfying in application of bacterial cultivation but also set the new state-of-the-art (SOTA) for growth-media prediction. On QM9, GraMixC achieves an MAE of 0.008, nearly matching the SOTA (w/o extra training data) of 0.007 (Fang et al., 2022), and represents the best result among non-GNN models.

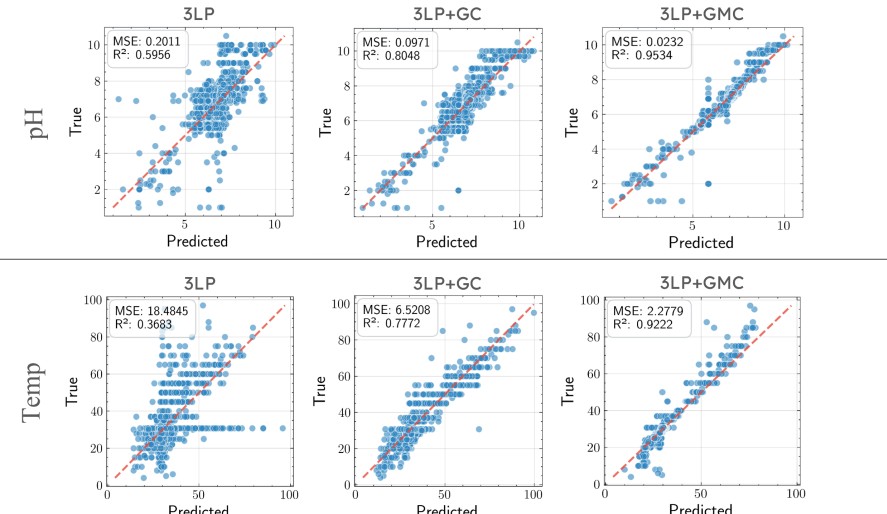

Figure 6: Illustration of the regression performance improvement example in 3LP by adding GC or GMC. Each column plots predicted vs. actual pH (top) or temperature (bottom). 3LP+GC (middle) outperforms the 3LP baseline (left), while 3LP+GMC (right) further boosts $R^2$ up to $> 0.9$.

**Number of configurations used.** We ablate the number of configuration levels in GMC. Fig. 7 shows that more configurations generally decreases MSE and increases $R^2$, confirming the value of multi-resolution information. Importantly, GMC often needs more than half as many total configurations to outperform GC, and performance plateaus—or even slightly declines—when including the last few configurations. These aligns with Pitsianis et al. (2023), who report a finite set of optimal configurations rather than continuous gains at infinite resolutions. Using all configurations available is still preferred.

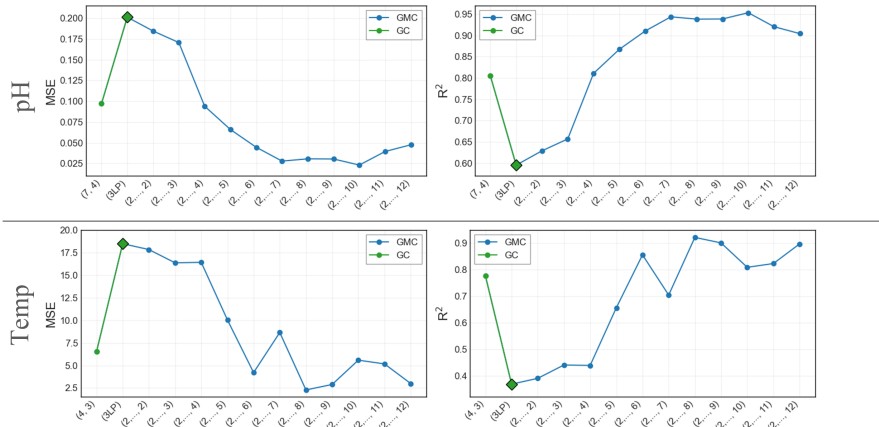

Figure 7: Ablation study on the number of configurations used on DSNI. On the blue curves (GMC), $[2, \ldots, i]$ denote fusing configurations from 2 through $i$ via GraMixC. On the green curves (GC), $(i, j)$ denote the best train/test configuration pair used in static concatenation. Incrementally mixing configurations improves performance and outperforms static concatenation.

### 4.3 QUALITATIVE EVALUATION OF CONFIGURATIONS.

Our final experiment compares configurations against standard representation-extraction methods. As discussed in Section 1, configurations can be viewed as special unsupervised representation learning. Fig. 3 already shows their advantage over self-supervised register tokens. Here, we replace GC/GMC with PCA (Jolliffe & Cadima, 2016), UMAP (McInnes et al., 2020), and a vanilla autoencoder (AE), each embed into dimensions the same number of as our configurations. We visualize these embeddings on MNIST (Fig. 8a; additional views in Section F.2). Qualitatively, SG-t-SNE (the reduction step in GraMixC) yields more uniform, well-separated clusters that respect global kNN connectivity rather than forming hubs. Fig. 8b quantifies downstream classification accuracy, where GC and GMC strongly outperform PCA, UMAP, and AE given the same embedding budget. These results confirm that mixed configurations provide a more expressive yet compact representation for downstream tasks.

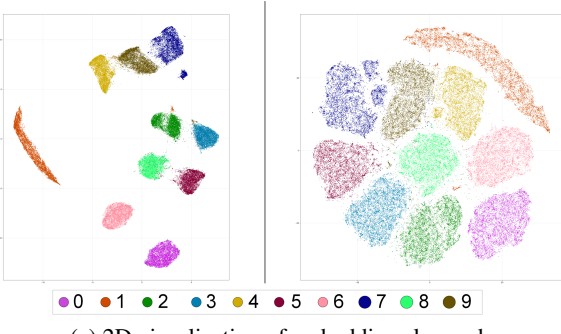

| | CE ↓ | Acc |
|---|---|---|
| 3LP+PCA | 0.157 | 0.971 |
| 3LP+UMAP | 0.181 | 0.975 |
| 3LP+AE | 0.158 | 0.969 |
| 3LP+GC | 0.046 | 0.992 |
| 3LP+GMC | **0.028** | **0.993** |

(a) 2D visualization of embeddings learned.      (b) Classification performance.

Figure 8: **(a)**: Illustration of 2D embeddings of MNIST using UMAP (left) and SG-t-SNE (right). **(b)**: Classification performance on MNIST using features from PCA, UMAP, autoencoder (AE), static configurations (GC), and GraMixC (GMC) at equal embedding dimensions. SG-t-SNE embeddings integrated via GC or GMC exploit multi-resolution structure to notably outperform other methods.

## 5 CONCLUSION

In this study, we investigate the functional mechanisms of configurations in downstream prediction tasks and identify three key properties. Based on this, we propose GraMixC, which dynamically mixes configurations through attention head. We apply it to the challenging task of 16S rRNA cultivation-media prediction task, and set a new state-of-the-art. Further validation across multiple standard tabular data benchmarks consistently reveals that GC (a static version of GraMixC) enhances baseline performance, while GraMixC demonstrates even more substantial improvements. Our results suggest that harnessing rich manifold priors via attention-driven fusion opens promising avenues for interpretable and robust learning in both scientific and conventional domains.

In future work, we plan to extend mixed configurations to more expressive networks and dynamically learn configuration alignment through end-to-end differentiable modules. Additionally, we will focus on exploring adaptive clustering for evolving data streams where train and test distributions may shift, which could further enhance the resilience of multi-resolution approaches.

ETHICS STATEMENT

We followed the ICLR Code of Ethics. All datasets (DSNI, NIH/DSMZ metadata, QM9, Boston Housing, MNIST, CIFAR-10) are publicly available and contain no personally identifiable information. Use complies with their licenses.

**Risks:** Our method predicts cultivation conditions from 16S rRNA features. Outputs are not lab-ready instructions and require expert validation. We discourage unsafe or unsupervised use, particularly with pathogenic organisms.

**Bias:** Training data reflect known biases (e.g., over-represented mesophiles). We report distributions (Section C) and evaluate across multiple benchmarks to reduce overfitting.

**Conflicts:** No competing financial interests. Experiments were run on institutional hardware.

REPRODUCIBILITY STATEMENT

Code, configs, and data-processing scripts are available at project-34CB and project-82CE on `https://anonymous.4open.science/r/project-34CB` and `https://anonymous.4open.science/r/project-82CE`.

All algorithms, hyperparameters, and dataset details are given in Sections C, 3.1, 3.2 and 4.1. Splits and seeds are fixed and provided. Figures and tables can be reproduced directly from the released scripts. Compute setup (GPU, runtime, nondeterminism) is documented in the README.

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

## A   LLM Usage Disclosure

We used GPT-5 only for writing support: editing text, drafting boilerplate (ethics/reproducibility), and LaTeX troubleshooting. It was not used for research ideas, experiments, or results. All technical content was created and verified by the authors. No sensitive data were shared.

## B   An Intuitive Example of Configuration Mixing

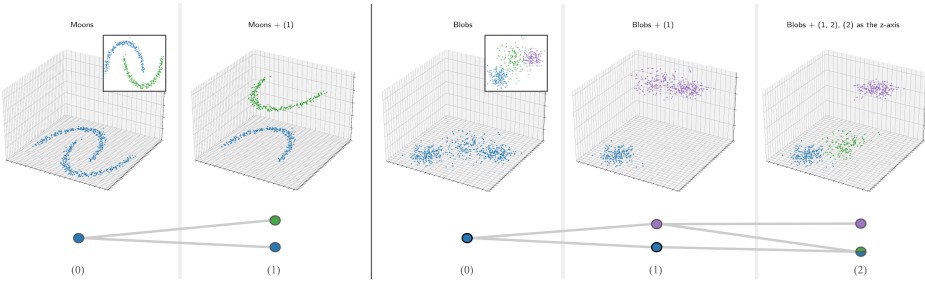

Figure 9: Illustration of multi-resolution clustering on synthetic datasets. GT is shown in the framed box in (0). Upper is the embedding of Moons (left) and Blobs (right) with corresponding configuration $(i)$ as third dimension; lower is the lineage diagram of the configurations.

To illustrate the necessity of fusing valid clusterings across resolution scales, we use two synthetic point-cloud datasets from scikit-learn: "Moons" and "Blobs." The Blobs dataset is tuned so that no

single clustering resolution recovers all three clusters. Fig. 9 visualizes each dataset in 3D, using the third axis to encode cluster assignments for the corresponding configuration: coarser configuration (1) and finer configuration (2). Configuration (1), by lifting some dots above the plane, cleanly separates the two Moon arcs but merges two (purple and green) of the Blobs clusters. Configuration (2), by itself, fails the Blobs with a different merge (blue and green). Only by fusing both configurations can all clusters be disentangled—the purple dots in (1) that falls down in (2), emerges correct as the green cluster. This toy example shows that multi-resolution clusterings alone are insufficient without a fusion mechanism. Our GraMixC use attention-based fusion to integrate these scales. While just one demonstration, it highlights the broader advantage of mixing configurations in complex settings.

## C  DSNI DATASET DISTRIBUTION

Fig. 10 shows the pH and temperature target distributions for DSNI across training and test splits. Both splits cover similar ranges, though with natural imbalance (e.g., mesophilic temperatures, pH 6–8) reflecting biases in the underlying NIH/DSMZ data. These distributions are important for interpreting regression performance and highlight potential challenges under distributional shift.

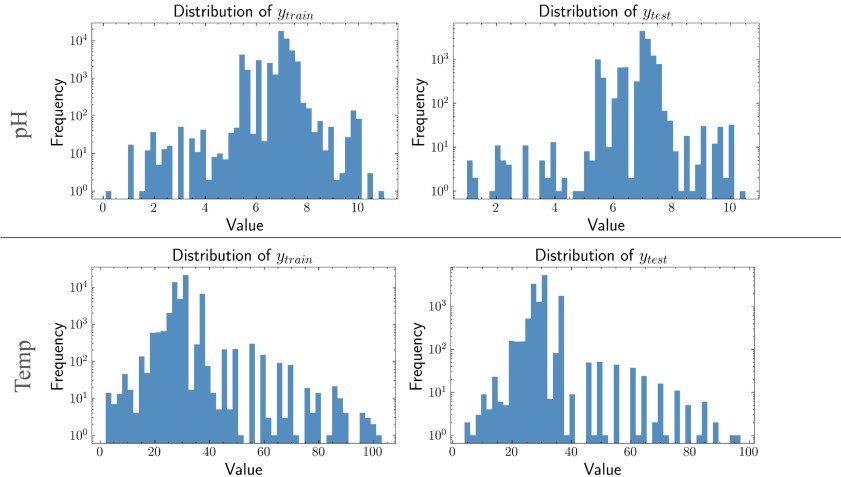

Figure 10: Illustration of target value distributions across train-test splits in DSNI dataset. The first row represents pH distributions and the second row represents temperature distributions. The first column represents the training set ($y_{\text{train}}$) and the second column represents the test set ($y_{\text{test}}$).

## D  SYNTHETIC CLUSTERING BENCHMARKS

In this section, we further discuss the limitations of conventional clustering methods raised in Section 3.1. We compare our modularity-based clustering strategy, which is used as the unsupervised layer in GraMixC, against widely-used clustering algorithms on synthetic 2D datasets.

Each row in Fig. 11 presents a distinct synthetic dataset distribution, ranging from custom-designed to standard scikit-learn datasets, including *Taiji*, spirals, circles, moons, varied blobs, anisotropy, blobs, and isotropic noise. Each column represents the result of one clustering method, annotated with Adjusted Rand Index (ARI) and execution time.

Unlike traditional clustering methods, the approach we adopted (last column: Modularity, implemented via kNN graph + Leiden community detection) consistently uncovers the underlying structure—even in challenging cases involving non-convex geometries, anisotropic spreads, or uneven density distributions. This comparison underscores the reliability and manifold sensitivity of our unsupervised segmentation approach, even before introducing multi-resolution fusion or downstream learning tasks.

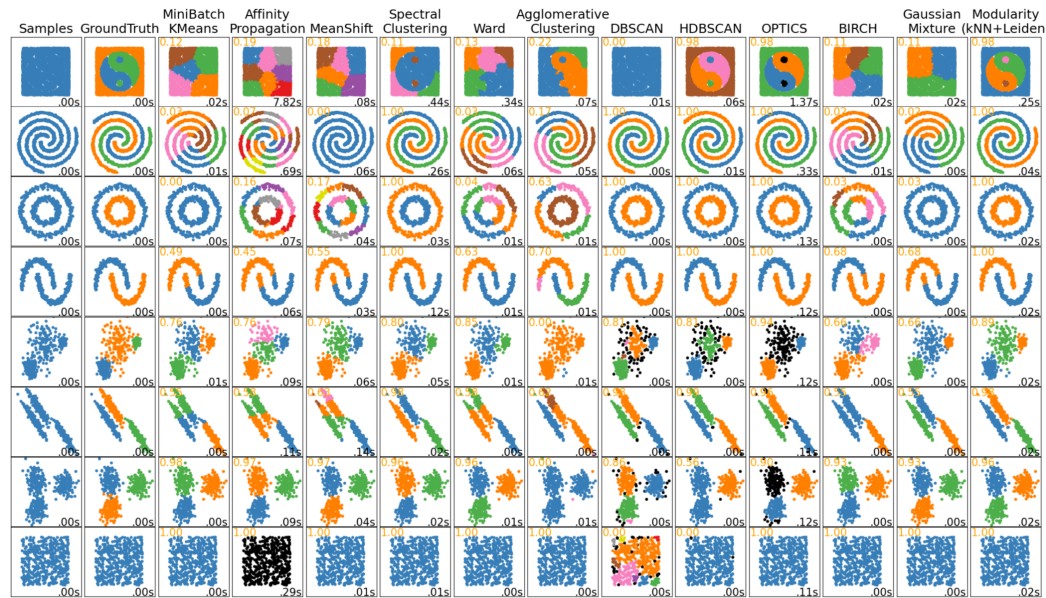

Figure 11: Illustration of clustering methods comparisons across multiple synthetic datasets. Rows correspond to different 2D point clouds—the first row is custom, others are from scikit-learn. Each method's result is labeled with ARI (top-left in yellow) and execution time (bottom-right in black). *Modularity: kNN+Leiden* (far right) accurately recovers ground-truth structures across different shapes and densities, with robustness to noise, anisotropy, and distribution variation.

## E RMS ALIGNMENT DETAILS

In Section 3.2 we introduced the Reverse Merge & Split (RMS) procedure for aligning multi-resolution configurations between train and test sets. Below we provide the full pseudo-code in Algorithm 1, using the same notation as the main text.

**Implementation notes.**

- We set $\theta = 0.1$ and compute ARI as in Hubert & Arabie (1985).
- We use 0.1 % of the train samples as anchors to form $\mathcal{A}$.
- The greedy matching loops over each train configuration $\omega_i$ to find its best-scoring test partner $\omega_j$, applies the label mapping, and removes both from further consideration to ensure one-to-one alignment.

The details for SCORE and $L_{\text{tw}}$ are covered in Algorithm 1 so we skip them here.

## F ADDITIONAL EXPERIMENTAL RESULTS

In Section 4 we introduced our experimental setup and high-level results. Here, we provide the full details and qualitative analyses that couldn't fit into the main body, including:

- Downstream task performance on three other benchmarks.
- Qualitative illustration of prediction versus true value on the three tabular baseline models.
- Embeddings from PCA and AE.

### F.1 ADDITIONAL EVALUATION OF PROPOSED MODULE

Table 2 extends our evaluation to three additional benchmarks: Boston Housing (regression), MNIST and CIFAR-10 (classification). We compare classical ensembles (RF, XGBoost, CatBoost), a 3-layer

---

**Algorithm 1** Reverse Merge & Split (RMS) Alignment

---

**Require:** $\boldsymbol{\Omega}_{\text{train}} \in \mathbb{N}^{N \times m_t}$, $\boldsymbol{\Omega}_{\text{test}} \in \mathbb{N}^{N \times m_s}$, anchor indices $\mathcal{A} \subset \{1, \ldots, N\}$, $\theta$
**Ensure:** Aligned $\boldsymbol{\Omega}_{\text{test}}$
1: $\mathbb{U} \leftarrow \{1, \ldots, m_t\}, \quad \mathbb{V} \leftarrow \{1, \ldots, m_s\}$
2: **for** $i$ in $\mathbb{U}$ **do**                      ▷ for each train configuration $\boldsymbol{\omega}_i$
3:     best_score $\leftarrow -\infty$, best_j $\leftarrow$ null
4:     $\boldsymbol{\omega}_i \leftarrow \boldsymbol{\Omega}_{\text{train}}[\mathcal{A}, i]$
5:     **for** $j$ in $\mathbb{V}$ **do**                 ▷ find best test configuration $\boldsymbol{\omega}_j$
6:         $\boldsymbol{\omega}_j \leftarrow \boldsymbol{\Omega}_{\text{test}}[\mathcal{A}, j]$
7:         $s \leftarrow \text{SCORE}(\boldsymbol{\omega}_i, \boldsymbol{\omega}_j, \theta)$
8:         **if** $s >$ best_score **then**
9:             best_score $\leftarrow s$, best_j $\leftarrow j$
10:         **end if**
11:     **end for**
12:     $M \leftarrow \text{PAIR\_MAPPING}(\boldsymbol{\Omega}_{\text{train}}[:, i], \boldsymbol{\Omega}_{\text{test}}[:, \text{best\_j}])$
13:     **for** $p = 1$ **to** $N$ **do**
14:         $\boldsymbol{\Omega}_{\text{test}}[p, \text{best\_j}] \leftarrow M(\boldsymbol{\Omega}_{\text{test}}[p, \text{best\_j}])$
15:     **end for**
16:     Remove $i$ from $\mathbb{U}$, remove best_j from $\mathbb{V}$
17: **end for**
18: **return** $\boldsymbol{\Omega}_{\text{test}}$

19: **function** PAIR_MAPPING($\boldsymbol{\omega}_i, \boldsymbol{\omega}_j$)
20:     $n_i \leftarrow \|\boldsymbol{\omega}_i\|_\infty, \quad n_j \leftarrow \|\boldsymbol{\omega}_j\|_\infty$
21:     **for** $p = 1$ **to** $N$ **do**              ▷ build confusion matrix $\boldsymbol{C} \in \mathbb{N}^{n_i \times n_j}$
22:         $\boldsymbol{C}[\boldsymbol{\omega}_i[p], \boldsymbol{\omega}_j[p]] \mathrel{+}= 1$
23:     **end for**
24:     Construct two-walk Laplacian $\boldsymbol{L}_{\text{tw}}$
25:     $\mathcal{F} \leftarrow$ Fiedler vector of $\boldsymbol{L}_{\text{tw}}$
26:     Split $\mathcal{F} \rightarrow (\mathcal{F}_i \in \mathbb{R}^{n_i}, \mathcal{F}_j \in \mathbb{R}^{n_j})$
27:     $\pi_i \leftarrow \text{argsort}(\mathcal{F}_i), \pi_j \leftarrow \text{argsort}(\mathcal{F}_j)$
28:     **return** mapping $k \mapsto \pi_i[\pi_j^{-1}(k)]$ for $k = 1, \ldots, \min(n_i, n_j)$
29: **end function**

---

MLP (3LP), and three neural tabular architectures (TabNet, TabTransformer, FT-Transformer) in three modes: baseline, static configuration concatenation (GC), and attention-based fusion (GMC).

Across almost all models and datasets, GC consistently improves performance over the raw baselines, and GMC provides further gains.

The sole exception is TabTransformer on Boston Housing, where GC yields only a marginal $R^2$ increase (0.811→0.813), but GMC degrades it (to 0.671), suggesting that attention-based fusion may disrupt already well-structured features in this case.

On MNIST, GC lifts accuracy above 99%, and GMC pushes it to 99.3–99.5%. On CIFAR-10, GC delivers dramatic gains (e.g. TabTransformer from 46.3% to 87.6%), and GMC further improves all models, with FT-Transformer+GMC reaching 95.5% accuracy. These results underscore that configuration integration via GraMixC is broadly effective, with only one minor counterexample.

### F.2 ADDITIONAL QUALITATIVE EVALUATION OF CONFIGURATIONS

In Section 4.3 we provided the embedding of MNIST digits using UMAP and SG-t-SNE (Fig. 8a). Here we provide the missing illustration of embedding with PCA and autoencoder (AE) in Fig. 12. As expected, they do not provide representation with clusters as separated as the former two methods.

With the final figure (Fig. 13) we visualize predicted vs. actual values from the tabular baselines on DSNI, filling in what is missing from Fig. 6.

Table 2: Regression/classification performance on Boston Housing (BHouse), MNIST, and CIFAR10.

| Dataset | BHouse | | MNIST | | CIFAR10 | |
|---|---|---|---|---|---|---|
| Metric | MSE ↓ | $R^2$ | CE ↓ | Acc | CE ↓ | Acc |
| RF | 0.022 | 0.884 | 0.247 | 0.969 | 1.681 | 0.463 |
| XGBoost | 0.022 | 0.881 | 0.066 | 0.980 | 1.296 | 0.539 |
| CatBoost | 0.016 | 0.913 | 0.096 | 0.975 | 1.230 | 0.567 |
| 3LP | 0.023 | 0.879 | 0.141 | 0.970 | 1.428 | 0.524 |
| 3LP+GC | 0.022 | 0.882 | 0.046 | 0.992 | 0.480 | 0.844 |
| 3LP+GMC | **0.017** | **0.909** | **0.028** | **0.993** | **0.220** | **0.949** |
| TabN | 0.033 | 0.822 | 0.130 | 0.964 | 1.499 | 0.463 |
| TabN+GC | 0.021 | 0.888 | 0.225 | 0.941 | 0.377 | 0.876 |
| TabN+GMC | **0.012** | **0.936** | **0.017** | **0.995** | **0.077** | **0.978** |
| TabT | 0.035 | 0.811 | 0.192 | 0.980 | 1.028 | 0.706 |
| TabT+GC | **0.035** | **0.813** | 0.040 | 0.993 | 1.049 | 0.704 |
| TabT+GMC | 0.061 | 0.671 | **0.018** | **0.994** | **0.458** | **0.911** |
| FTT | 0.032 | 0.826 | 0.098 | 0.980 | 0.415 | 0.874 |
| FTT+GC | 0.030 | 0.838 | 0.029 | 0.993 | 0.437 | 0.870 |
| FTT+GMC | **0.026** | **0.860** | **0.018** | **0.995** | **0.157** | **0.955** |

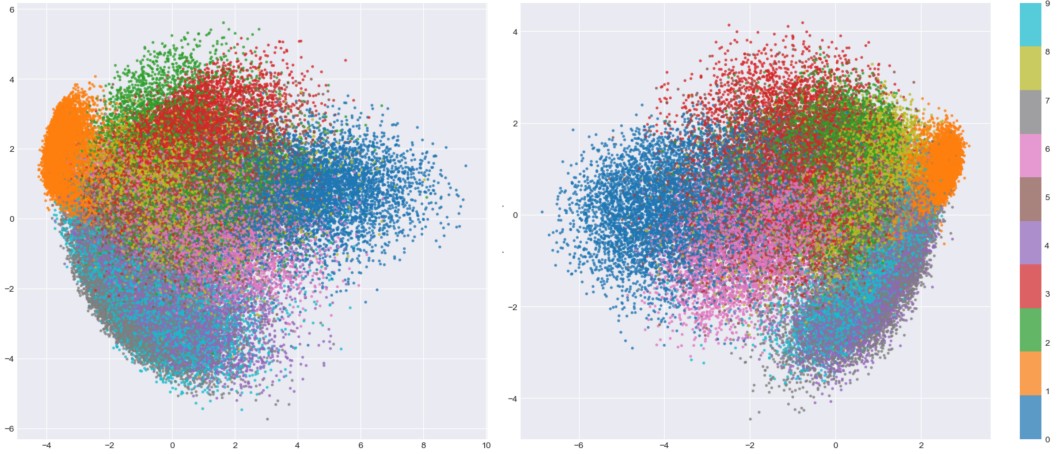

Figure 12: Illustration of 2D embeddings learned by PCA (left) and AE (right) on MNIST.

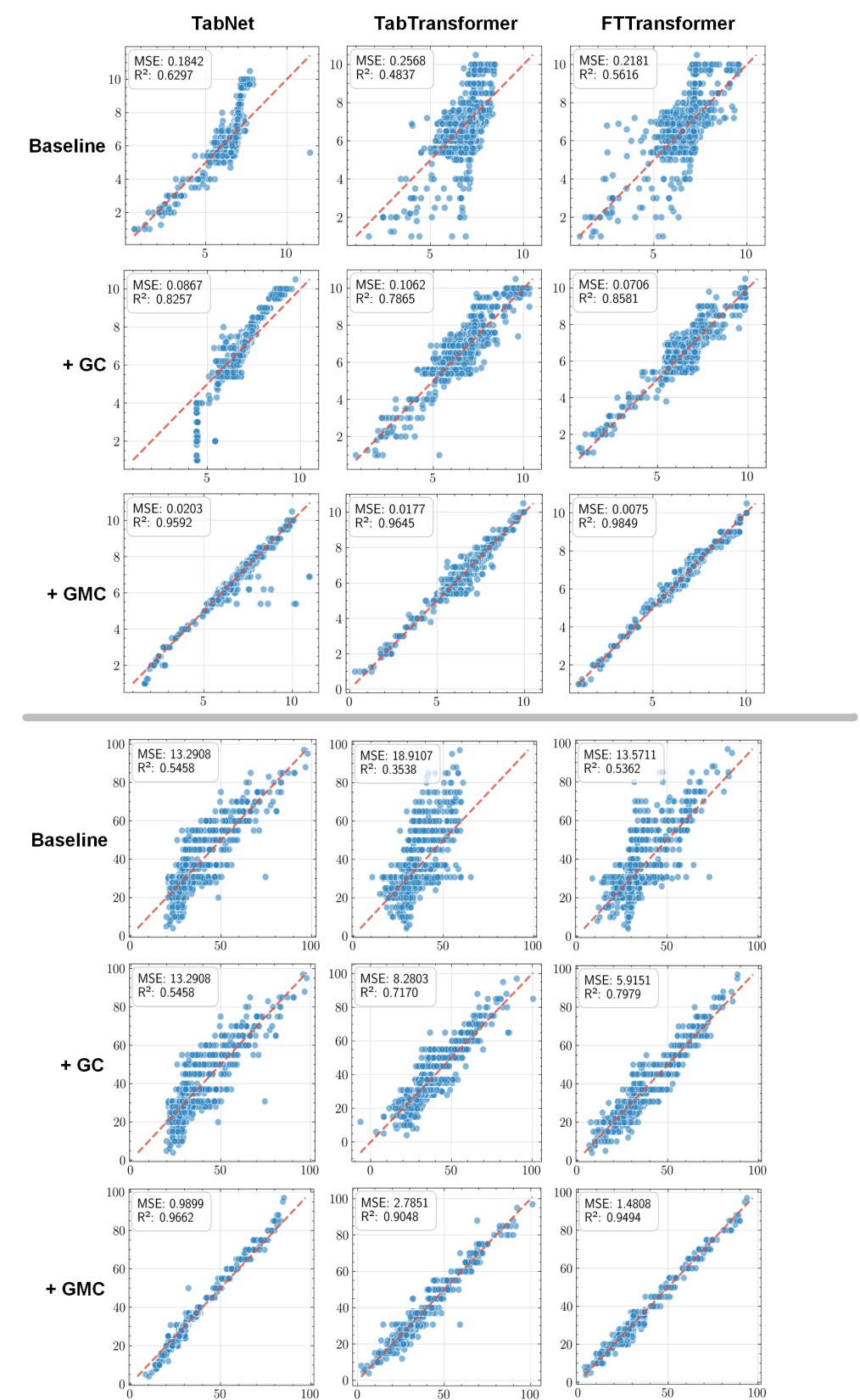

Figure 13: Illustration of the regression performance improvement example in TabNet, TabTransformer and FT-Transformer by adding GC or GMC. Each plots predicted vs. actual value.

