# OpenReview forum: "Mixing Configurations for Downstream Prediction"
_ICLR.cc/2026/Conference — ICLR 2026 Conference Withdrawn Submission_

### Official Review · Reviewer_2yMt · 2025-10-29

**Soundness:** 2
**Presentation:** 1
**Contribution:** 2
**Rating:** 2
**Confidence:** 2

**Summary:**

The authors propose GraMixC, an approach for combining multiple hierarchical clusterings of a dataset for incorporation as features in a downstream predictor. The GraMixC method first generates multi-resolution cluster memberships using unsupervised learning methods, and then aligns these clusterings using their proposed Reverse Merge & Split method. The clusterings are then combined using attention heads, and integrated into downstream prediction. The authors demonstrate their approach on several datasets and find that incorporation of the cluster features boosts performance when added to existing deep tabular models.

**Strengths:**

- Despite clarity concerns noted below, the paper proposes an interesting and useful paradigm for integrating clustering results into downstream prediction models.
- Existing experiments appear to be well executed with a good mix of synthetic, benchmark, and realistic datasets. Performance improvements seem strong, although need to be properly contextualized.

**Weaknesses:**

This paper is written with insufficient clarity for the generalist ICLR audience, in my opinion. Overall, the inaccessibility of the manuscript prevents a fully informed review that is reflected in my confidence score. Detailed issues include:

- The paper assumes deep familiarity with community detection literature (BlueRed Front, modularity-based clustering, parallel-DT) without providing a background or related works section. For example, the term "configuration" is only defined in Section 3.1 when it is central to understanding the paper. Similarly, the distinction between "valid" and "pseudo" configurations appears very late despite being referenced several times prior. Overall, this pattern of retroactively defining terms causes confusion.

- I found the preliminary results section to be interesting but abrupt in its placement. Reading the paper chronologically, its not clear how the configurations were even generated by this point in the paper.

- The experiments lack comparison to alternative approaches. For example, no comparison is made to DeepCluster despite the fact that it also uses clustering for downstream tasks.

- In general, there is limited discussion of how GraMixC differs from other hierarchical or multi-resolution clustering methods in deep learning. This hinders readers from understanding the novelty of GraMixC's contributions. The manuscript would benefit greatly from a related works section.

- Several ablations could be performed to better empirically motivate methodological choices. For example:
  - Could fixed-γ Leiden replace BlueRed+parallel-DT?
  - The necessity of register tokens is mentioned (page 3) but never ablated
  - No comparison of GraMixC to more naive approaches for featurizing hierarchical clusterings.

**Questions:**

I would appreciate seeing the experimental comparisons suggested above to demonstrate that GraMixC offers improvement upon more naive approaches.

---

### Official Review · Reviewer_QaB6 · 2025-10-29

**Soundness:** 2
**Presentation:** 1
**Contribution:** 2
**Rating:** 2
**Confidence:** 4

**Summary:**

This paper introduces a method for improving downstream prediction by using a form of multiscale clustering. The idea is to first extract configurations (hierarchical groupings of samples) learned from graphs. Then, align the training and test configurations using a proposed procedure, and finally fuse via attention heads trained for the prediction task. The authors evaluate their method on several datasets, including a biological task and a set of tabular benchmarks. Reported results show consistent improvements.

**Strengths:**

The authors demonstrate that the method can help improve the performance of several prediction models on multiple datasets.
The concept of using unsupervised learning to improve supervised tasks is interesting (although not new).

**Weaknesses:**

The paper is poorly written, hard to follow, and many details about the evaluations are unclear.
The motivation isn’t explained, and the flow of the abstract and intro is bad. The paper's main message is convoluted and poorly conveyed.
It is also not clear what in the paper is new, and what is a combination of existing methods.

The authors don’t explain the computational overhead of this method compared with simply applying the downstream classifier/regressor. Same for training.
The training protocol is not clear, so it's hard to judge the results. Specifically, how are hyperparameters tuned? Even the number of configurations dramatically affects the results.
There is no ablation (evaluating the effect of varying a parameter isn’t an ablation). The GC is an ablation for the attention, but other parts are not ablated.

**Questions:**

here are no real comparisons to existing methods; the comparison to tsne, pca, ae isn’t really aligned with the paper's goal. The paper isn’t presenting a dimensionality reduction method for learning these features, so why does it make sense to compare to DR methods?

What is the point of comparing UMAP to t-SNE using your method? Clearly, these methods generate different types of embeddings, and this visualization does not support the claims made in the paper. Additionally, the hyperparameters of t-SNE could also influence this property.
What’s the added value of figure 3 in the main text? The metrics are already presented in the paper

---

### Official Review · Reviewer_mKFU · 2025-11-01

**Soundness:** 2
**Presentation:** 2
**Contribution:** 2
**Rating:** 6
**Confidence:** 2

**Summary:**

This paper studies the role of multi-resolution clustering configurations for improving downstream prediction. The authors introduce GraMixC, a plug-and-play module that extracts hierarchical configurations from unsupervised or self-supervised clustering, aligns them via a new Reverse Merge & Split (RMS) technique, and fuses them through attention heads before passing them to downstream predictors. The approach is evaluated on 16S rRNA cultivation-media prediction task (DSNI dataset) and several standard tabular and vision benchmarks. GraMixC improves regression and classification performance across baselines such as 3-layer MLPs, TabNet, TabTransformer, and FT-Transformer, achieving state-of-the-art results on DSNI with R² up to 0.98.

**Strengths:**

- The paper is clear about its motivation with sufficient significance and quality.
- The paper introduces a creative idea, mixing multi-resolution cluster configurations to capture global manifold structure in data.
- GraMixC can be easily attached to existing predictors, which makes it broadly applicable.
- The alignment procedure (RMS) and the configuration formulation are mathematically detailed, showing rigor behind the method.
- The attention maps, ablation studies, and qualitative examples provide good intuition about why the approach works.
- The application to microbiome data (16S rRNA) shows strong practical potential in scientific domains.

**Weaknesses:**

- The paper is quite dense. Many sections, especially those describing the clustering and RMS steps, are mathematically overloaded and could use higher-level intuition or easy separation for better understanding.
- The method largely builds upon existing clustering and attention mechanisms; the main novelty mainly lies in combining and aligning them.
- The paper does not clearly quantify the computational overhead of multi-resolution clustering and RMS alignment compared to single-resolution or embedding-based alternatives, i.e no computational complexity clearly mentioned for the method. How does it scale to larger dataset?
- The connection between “register tokens” in vision transformers and configurations is interesting but underexplained. The analogy could be more explicitly tied to the main contribution.
- Especially for multimodal settings, some comparisons (e.g., contrastive or adapter-based models) would help position the work better.
- Missing clear ablation studies regarding specific hyperparameters choice (e.g $\theta$ and $k$).

**Questions:**

- How sensitive is GraMixC to the number of configurations? Do too many resolutions introduce redundancy or instability?
- Could RMS alignment be learned end-to-end instead of computed heuristically?
- What is the runtime overhead of GraMixC compared to the base models, especially on large datasets?
- How generalizable is the method to high-dimensional continuous embeddings where discrete clustering is less natural?
- What is the AE configuration you used for fig 8(b) performance?
- What is the performance of GraMixC on any imbalanced datasets where some clusters are much smaller?

---

### Official Review · Reviewer_Q9qr · 2025-11-02

**Soundness:** 2
**Presentation:** 3
**Contribution:** 2
**Rating:** 2
**Confidence:** 3

**Summary:**

The paper introduces a hierarchical configuration learning module GraMixC as part of a prediction pipeline. It is evaluated on a bacterial species prediction task. The authors provide extensive experimentation and evaluation procedure

**Strengths:**

-	The proposed approach is novel and is well presented
-	The evaluation experiments are extensive and well documented
-	Performance on the bacterial species prediction task is demonstrably improved

**Weaknesses:**

-	The proposed hierarchical configuration learning task is not well motivated or defined.  The evaluation protocol does not seem to be directly corresponding to the stated task of configuration learning. The main example is in predicting bacterial species. Perhaps the whole approach is a better method for the specific domain rather than a general-purpose method.
-	The module seems to be evaluated in a somewhat ad hoc manner. For example, in Figure 1, bird is clustered with horse and deer but cat and dog are clustered separately from them at configuration level 1 (and from is on its own). Yet, cat and dog are closer to the horse and deer than a bird and most humans, even kids, would cluster those together (although language plays an important role in the process of categorization  https://doi.org/10.1002/wcs.96). Biology in general provides a great ground truth for evaluation of hierarchical similarity for example, using taxonomical hierarchy or species mimicry and there are plenty of benchmark datasets
-	The entire set of evaluation metrics is somewhat puzzling. For example, it is unclear how attention maps are relevant to the method’s evaluation. In general, attention maps have been shown not to be particularly great about learning fine grain features. I am not sure what the authors are trying to show with the attention maps.

**Questions:**

Can you clearly state and motivate the task/problem you are proposing? Can you reframe the evaluation procedure to fit the stated objective a priori?

---

### Note · Authors · 2025-11-12

I have read and agree with the venue's withdrawal policy on behalf of myself and my co-authors.